# Applications of a Standardized Green Tea Catechin Preparation for Viral Warts and Human Papilloma Virus-Related and Unrelated Cancers

**DOI:** 10.3390/molecules25112588

**Published:** 2020-06-02

**Authors:** Noriyuki Miyoshi, Hiroki Tanabe, Takuji Suzuki, Koichi Saeki, Yukihiko Hara

**Affiliations:** 1Graduate School of Integrated Pharmaceutical and Nutritional Sciences, University of Shizuoka, Shizuoka 422-8526, Japan; 2Faculty of Health and Welfare Science, Nayoro City University, Nayoro, Hokkaido 096-8641, Japan; htanabe@nayoro.ac.jp; 3Faculty of Agriculture, Yamagata University, Yamagata 990-8560, Japan; taksuzuk@e.yamagata-u.ac.jp; 4Regenerative Medicine iPS Gateway Center Co., Ltd., Tokyo 150-0012, Japan; ks.saeki@gmail.com; 5Tea Solutions, Hara Office Inc., Tokyo 130-0012, Japan

**Keywords:** green tea catechins, epigallocatechin gallate, human papillomavirus, gene expression

## Abstract

Most cell-based and animal experiments have shown that green tea catechins (GTC) exhibit various health benefits. In human experimental and epidemiological studies, there are conflicting results, and more precise investigations are required. One of the most effective ways to prove beneficial health effects in humans might be clinical intervention studies. Polyphenon^®^E was developed as a standardized GTC preparation, which was approved by Food and Drug Administration of US in 2006 as a medication to treat genital warts (Veregen^®^ or sinecatechins). Positive efficacy of Polyphenon^®^E/sinecatechins/Veregen^®^ (PSV) on anogenital warts has been demonstrated in several epidemiological studies and there have been several case reports to show the clinical effectiveness of PSV. In addition, several studies have provided evidence to suggest that PSV is effective in other human papillomavirus (HPV)-related diseases, although some studies failed to show such effects. Since (−)-epigallocatechin gallate (EGCG) is the major component of PSV, the mechanism of the action of PSV might be deduced from that of EGCG. The microarray analysis of the biopsy samples from the patients suggested that apoptosis induction and the downregulation of inflammation are involved in the mechanism of the action of PSV in the clearance of anogenital warts. Cell-based and animal experiments using PSV also demonstrated effects similar to those elicited by EGCG, explaining how PSV works to induce apoptosis and exert anti-inflammatory actions in HPV-related diseases. Future studies would clarify what kinds of diseases respond effectively to PSV, showing health benefits of GTC and EGCG in humans.

## 1. Introduction

Green tea is made from the leaves and buds of the *Camellia sinensis* (Theaceae) plant and it has been shown to have health benefits on many diseases, such as cancer, obesity, diabetes, and cardiovascular, neurodegenerative, and microbial diseases [1,2,3]. A number of human epidemiological studies have provided evidence for the beneficial health effects of consumption of green tea, although there are conflicting results. One of the most effective ways to prove beneficial health effects in humans might be clinical intervention studies.

Green tea leaves contain various polyphenolic compounds, including (−)-epicatechin, (−)-epigallocatechin, (−)-epicatechin gallate, and (−)-epigallocatechin gallate (EGCG, Figure 1). These catechins are purified from green tea leaves by water extraction, subsequent ethyl acetate extraction, and then by column chromatography using water/alcohol as an eluant. Mitsui Norin Co. Ltd. in Japan established the method to manufacture a standardized preparation of green tea catechins named Polyphenon^®^E. Polyphenon^®^E is composed of EGCG (>65%), (−)-epicatechin (>10%), (−)-epicatechin gallate (<10%), and (−)-epigallocatechin (<10%) with minor catechin derivatives and non-catechin compounds (<10%) [1]. 

In the initial and subsequent trials at the Beijing Cancer Center Hospital in 1990 in China, the application of Polyphenon^®^E ointment on genital warts (*Condyloma acuminata*) was found to successfully eliminate the warts [1]. Based on these findings, MediGene AG in Germany carried out clinical phase II/III trials internationally. In 2006, the United States Food and Drug Administration (FDA) approved the marketing of the sinecatechins ointment as a prescribed Botanical Drug in USA [1], which has been marketed in EU countries and USA under the trademark of Veregen^®^ [1]. FDA approved the topical treatment of external genital warts (EGW) and perianal warts with sinecatechins 15% in immuocompetent patients from the age of 18 years for three times daily until complete the clearance of warts or for up to 16 weeks [4].

It should be noted that in the years preceding 2018, FDA has received over 800 botanical investigational new drug applications and pre-investigational new drug applications meeting requests and only two botanical new drug applications have been approved in the U.S.: Veregen^®^ in 2006 and Fulyzaq^®^ in 2012 [5].

Polyphenon^®^E/sinecatechins/Veregen^®^ (PSV) appears to be useful for human intervention studies, since the potential efficacy of PSV has been demonstrated in several literatures [6,7,8,9]. In addition to EGW caused by human papilloma virus (HPV), PSV might have promising effects on HPV-associated other diseases. HPV can cause malignancies (cervical, penile, vulvar, vaginal, anal, and oropharyngeal) and noncancerous cutaneous manifestations after infecting the basal cells of the epithelial mucosa. Noncancerous cutaneous manifestations include common warts (verruca vulgaris), plantar warts, plane warts, genital warts (GW), and AGW [10]. 

A number of studies of PSV’s effects in these HPV-related diseases have been done since the last reviewing [1]. This review article provides up-dated information on these subjects, as examined by human studies and discusses mechanistic aspects of the action of PSV.

## 2. The Efficacy of PSV on GW and Proactive Sequential Therapy after Ablative Treatments

In an early stage of epidemiological studies of PSV, a randomized, double-blind, placebo-controlled study in which 125 male and 117 female were participated found that treatment with 15% Polyphenon^®^E ointment resulted in a statistically significant difference in the complete clearance of all baseline EGW (61.0% in males and 56.8% in females) when compared to placebo (40.5% in males and 34.1% in females) [6]. In the case of 10% Polyphenon^®^E cream, complete clearance was found in 53.8% of males and 39.5% of females.

In a clinical study involving 503 patients, Stockfleth et al. [7] found that the complete clearance of all baseline and new anogenital warts (AGW) was about 53% of patients treated with Polyphenon^®^E 15% ointment, 51% of those treated with Polyphenon^®^E 10% ointment, and 37% for placebo group. In a clinical trial in which 502 male and female patients aged ≥18 years were received application of sinecatechins ointment 15% or 10% or placebo three times daily for a maximum of 16 weeks or until complete clearance, Tatti et al. [8] found that complete clearance of all baseline and new warts was achieved in 57.2% and 56.3% of patients that were treated with 15% and 10% ointments, respectively, as compared to 33.7% for placebo.

A systematic review and meta-analysis of these three clinical trials showed that treatments with Polyphenon^®^E 15% and 10% ointments resulted in significantly higher rates of complete clearance of baseline and new warts when compared with controls with very low recurrence rates [9].

Several recent literatures conducting a systematic review and meta-analysis have shown the effectiveness of PSV [11,12] similar to the previous one [9]. In addition, a recent retrospective cohort study on 24 children with AGW and/or EGW without a control group for comparison revealed the efficacy of sinecatechins ointment as a promising therapy in children. The major findings are as follows: (i) mean age at treatment initiation was 8.0 years; (ii) the range of median duration of warts at treatment initiation was 0.09−12.62 years; (iii) 16 patients (66.7%) had a reduction in the number and/or size of the warts and four patients (16.7%) had complete resolution; (iv) the range of median treatment duration was 0.6−21.8 months; and, (v) median time to complete resolution was 2.9 months (range 1.3–7.7) [13].

A combination of a known therapy with PSV might be more effective for EGW. On et al. [14] conducted single-blinded randomized controlled study in which 42 subjects with at least two EGW lesions were treated for cryotherapy and then with or without catechins 15% ointment twice daily after cryotherapy. After 16 weeks of treatment, they found a significant reduction (−5.0) in mean number of lesions from baseline in the cryotherapy-sinecatechins ointment group when compared to cryotherapy alone (−2.1), suggesting the better efficacy of cryotherapy plus sinecatechins 15% ointment than cryotherapy alone in the reduction of EGW. 

Juhl et al. [15] conducted a retrospective review of 27 patients in whom combined therapy for EGW was performed while using one or two sessions of cryotherapy combined with an anti-GW drug 25% podophyllin, and topical application of sinecatechins 15% ointment. The result indicated excellent initial clearance rate of 96.3% with a relative risk (RR) of 7.4% during six months of follow-up [16].

Similarly, Puviani et al. [17] evaluated the efficacy of sinecatechins 10% ointment applied twice daily with “difficult to treat” EGW after CO_2_ laser ablative treatment in a placebo-controlled trial on a total of 87 patients in whom mean baseline number of the lesions was 6.5. After three months, a total of 86 treated and untreated subjects who completed the study, the treated group had a RR of 5% as compared to that of 29% in the untreated group, which suggests that sinecatechins may be suitable in proactive sequential therapy after ablative strategies to lower RR.

## 3. PSV Effects on Other HPV-Related Diseases

As mentioned above, PSV has been demonstrated to be effective on GW and other viral warts. The causative agents are human papilloma viruses (HPVs), such as six, 11, and 16 types [16,18]. Several studies have examined the possible application of PSV to other diseases that are caused by HPVs. 

Ahn et al. [19] examined clinical efficacy of Polyphenon^®^E ointment or capsule or both in 51 patients with HPV-infected cervical lesions as compared with 39 untreated patients. The ointment was locally applied to 27 patients twice a week. For oral delivery, a 200 mg of Polyphenon^®^E or EGCG capsule was taken orally every day for eight to 12 weeks. As a result, clinical response was observed in 74%, 75%, 50%, and 60% of patients under therapy with the ointment alone, the ointment plus Polyphenon^®^E capsule, Polyphenon^®^E capsule alone, and EGCG capsule alone, respectively. Overall, a 69% response rate was attained for treatment with Polyphenon^®^E or EGCG, as compared with a 10% response rate in untreated controls, which suggests that Polyphenon^®^E ointment and capsule can be a potential therapy regimen for patients with HPV-infected cervical lesions. Together with the findings in a case-control study showing that green tea intake (OR = 0.551, 95% CI = 0.330–0.919) was identified as protective factors against cervical cancer or cervical intraepithelial neoplasias, green tea might reduce the risk of cervical cancer [20].

In contrast, a later study failed to confirm the result. Garcia et al. [21] conducted a randomized, double-blind, placebo-controlled trial of Polyphenon^®^E in 98 women with persistent HPV infection and low-grade cervical intraepithelial neoplasia to evaluate the potential of Polyphenon^®^E for cervical cancer prevention. The patients received either Polyphenon^®^E (containing 800 mg EGCG) or placebo once daily for four months. The results indicated that Polyphenon^®^E did not promote the clearance of the virus and intraepithelial neoplasia, suggesting that further studies are required in order to know the effect of Polyphenon^®^E on cervical cancer.

Anogenital verrucae (AV) are benign, HPV-induced tumors of the anogenital skin and mucosa. Chamseddin et al. [22] performed a retrospective review of 37 children under 12 years of age with AV treated with 5% cream of an anti-GW drug imiquimod and sinecatechins 15% ointment. The results indicated that imiquimod 5% cream and sinecatechins 15% ointment are both moderately effective in these preadolescent children, with a trend toward greater effectiveness of sinecatechins. Combination therapy with other treatments did not significantly increase the effectiveness of topical therapies without serious complications. 

## 4. Case Eeports of PSV Effects on Viral Warts and Other HPV-Related Diseases

Several case reports have been presented for PSV effects on viral warts [23,24,25,26,27] (Table 1) and other HPV-related diseases [28,29,30]. For example, one study found that topical treatment with a sinecatechins 10% ointment thrice daily for three weeks in a 34-year-old man with atopic dermatitis, who had not responded to 5-fluorouracil ointment, water filtered infrared A irradiation, and topical retinoids, resulted in a complete remission of all facial warts within 20 days [23]. These findings demonstrate the excellent efficacy of PSV on various kinds of warts without significant adverse effects, encouraging application of PSV to some types of HPV-mediated diseases.

Examples of other HPV-related diseases are as follows. Gupta et al. [28] reported a case of a 45-year-old female who showed a complete response to the treatment with Veregen^®^. The patient had high grade vulvar intraepithelial neoplasia and squamous cell carcinoma in situ and failed to show any clinical response, including topical Imiquimod 5% ointment. Treatment with Veregen^®^ resulted in the complete resolution of all lesions, with mild scarring on the vulva at the end of six weeks and biopsies of the scarred areas provided a negative result for dysplasia or carcinoma. Remission remained after 11 follow up months.

Topical Polyphenon^®^E 10% treatment could also be very effective in the treatment of “seborrheic keratosis-like” lesions of the inguinal area. In an immunocompetent 26-year-old Caucasian man who failed to respond to a two-month imiquimod treatment, all the lesions disappeared with only hyperchromic residues after three months of the Polyphenon^®^E 10% treatment [29].

In contrast, Henrickson et al. [30] reported that two brothers with recalcitrant verrucae did not respond to traditional topical therapies, including sinecatechins, liquid nitrogen, and imiquimod, although they had full response to topical 3% cidofovir. 

Future clinical studies would more clearly reveal the usefulness of PSV as a medication in some types of HPV-related diseases.

## 5. PSV Effects on HPV-Unrelated Cancers 

In a phase II trial in patients with asymptomatic, Rai stage 0 to II chronic lymphocytic leukemia, in which a total of 42 patients received Polyphenon^®^E at a dose of 2000 mg twice daily for up to six month, Shanafelt et al. [31] found that 31% of patients showed a sustained reduction of ≥20% in the absolute lymphocyte count and 69% experienced at least a 50% reduction in palpable lymphadenopathy.

In a phase II pharmacodynamic prevention trial, patients with a bladder tumor were randomized to receive Polyphenon^®^E containing either 800 or 1200 mg of EGCG or placebo for 14 to 28 days prior to transurethral resection of bladder tumor or cystectomy. Of 31 patients, 29 completed the study. In Polyphenon^®^E groups, tissue PCNA, a biomarker of cellular proliferation, and clusterin, which is up-regulated in many cancers, were dose-dependently downregulated. Observed downregulation of these tissue biomarkers suggests that Polyphonon^®^E might be useful in bladder cancer prevention [32].

Crew et al. [33] conducted a clinical study, in which women with stage I–III breast cancer who completed adjuvant treatment were randomized to Polyphenon^®^E 400 mg, 600 mg and 800 mg twice daily or matching placebo groups for six months and evaluated the Polyphenon^®^E (26 patients) and placebo groups (eight patients). At two months, the Polyphenon^®^E group had a significant decrease (−12.7%) in serum levels of hepatocyte growth factor, which is an important target in cancer drug development as compared to the placebo group (+6.3%). They also observed a trend toward a decrease in serum cholesterol with Polyphenon^®^E. The result suggests a potential role of Polyphenon^®^E in breast cancer. These findings imply the health benefits of PSV not only on HPV-related cancers, but also on other cancer types. 

On the other hand, there are several reports that fail to demonstrate the health benefits of PSV. The results of a randomized, double-blind, placebo-controlled trial that was conducted by Nguyen et al. [34] showed that daily consumption of Polyphenon^®^E 800 mg for three to six weeks gave favorable results, but not statistically significant changes in serum biomarker, such as prostate-specific antigen. Biomarkers, such as cell proliferation, apoptosis, and angiogenesis in the prostatectomy tissue, did not differ between the treatment arms. The results suggest that further study is required to know prostate cancer preventive activity of Polyphenon^®^E. A similar study that was conducted by Kumar et al. [35] for one year found that Polyphenon^®^E did not affect biomarkers of obesity in 97 men who were at high risk for prostate cancer.

## 6. Clinical Studies of PSV Effects on Other Diseases 

PSV might have therapeutic beneficial potential in some diseases. In a pilot study to evaluate the safety and efficacy of an oral dose of Polyphenon^®^ E in patients with mild to moderate ulcerative colitis, 20 patients were randomized to active therapy and placebo groups. Nineteen subjects received >1 dose of study medication (15 Polyphenon^®^E and four placebo groups). After 56 days of therapy, the response rate was higher in the Polyphenon^®^E group (66.7%) than the placebo group (0%). The corresponding remission rates of the former and the latter were 53.3% and 0%, respectively, which suggests that Polyphenon^®^E has a favorable effect in patients with ulcerative colitis [36].

However, a randomized, crossover, double-blind, placebo-controlled clinical trial in recessive dystrophic epidermolysis bullosa showed that Polyphenon^®^E treatment was not more effective than placebo [37]. In addition, phase I (10 patients) and phase II (12 patients) trials to assess the neuroprotective effects of Polyphenon^®^E in multiple sclerosis found that Polyphenon^®^E at a dose of 400 mg of EGCG twice a day is not futile at increasing brain levels of *N*-acetyl aspartate, which serves a marker of neuronal viability/number [38]. 

## 7. Inconsistency among Study Results and Adverse Effects of PSV

As discussed above, a number of results from clinical studies have provided evidence to show the efficacy of PSV on viral warts and some other diseases. However, several studies have failed to show such favorable effects [30,34,37,38]. These inconsistent results may have arisen from potential confounding factors, including differences in disease state at drug application, infected virus types, drug formulations, treatment methods, genetic factors, and life styles, such as cigarette smoking and alcohol drinking. Thus, future high quality randomized clinical trials would reveal the efficacy of PSV. 

Regarding the adverse effects of PSV, the clinical treatment with PSV was, in general, well tolerated and safe in most cases and caused mild to moderate side effects [39]. Erythema, pruritus, burning, and swelling were observed in 10–20% of patients [39].

In a recent retrospective cohort study on 24 children, adverse events were found to be light local irritation in seven patients (29.2%) [13]. A patient had some pain and discomfort at the application site but tolerated to complete resolution of vulvar carcinoma, according to a case report by Gupta et al. [28]. Rob et al. [26] reported their experience that local adverse reactions, such as burning and pain, are less common when compared with other topical treatments in the sinecatechins application to adult patients.

In a randomized, double-blind, placebo controlled clinical study with 24 receiving Polyphenon^®^E and 24 receiving placebo, nausea was the most common event, with an incidence rate of 16% [34]. Incidence rates of diarrhea (8%) and headache (4%) were less than those of placebo groups. One subject in the Polyphenon^®^E group had a mild liver transaminase elevation (4%) in blood. Similarly, a randomized placebo-controlled study reported one participant on Polyphenon^®^E treatment who had a serious hepatotoxicity with the transaminase levels elevated 15 times above normal and elevated bilirubin, probably due to a lot difference in the content of minor catechins [38]. 

## 8. Mechanistic Consideration

A number of cell-based and animal experiments have provided the molecular mechanism of EGCG’s action. EGCG is the major component of PSV, whereby the mechanism of actions of PSV may be reasonably deduced from those of EGCG. A comprehensive review conducted by Wang et al. [40] summarized the role of EGCG in prevention of human cervical cancer, in which HPVs are involved. Major actions include antiproliferation, apoptosis induction, anti-angiogenesis, and immunomodulation (Figure 2).

Microarray assays of the biopsy samples from AGW patients suggested that sinecatechins caused the downregulation of antiapoptotic and proinflammatry genes, and these modulations have been suggested to be the basic mechanism to be operated in clearance of GW and AGW treated with PSV [16,41,42,43,44,45].

EGCG can cause cell cycle arrest, leading to the inhibition of cell proliferation, apoptosis, and angiogenesis through the down-regulation of HPV oncogene E6, HPV oncogene E7, epidermal growth factor receptor (EGFR), protein kinase B/phosphoinositide-3-kinase (AKT/PI3K), mammalian target of rapamycin (mTOR), and extracellular signal-regulated kinase 1/2 (ERK1/2) (Figure 2). The EGCG-mediated suppression of E6 would up-regulate tumor suppressor proteins p53, p21, and p27, leading to the downregulation of cycline-dependent kinase (CDK2) to cause cell cycle arrest. The downregulation of E6 also causes upregulation of proapoptotic proteins caspase-8 and Bax [40]. EGCG-mediated upregulation of p53 causes the activation of proapoptotic protein Bak leading to apoptosis [16,40,46].

EGCG might elicit its activity through its antioxidant properties (Figure 2) [16]. ROS can activate nuclear factor-κB (NF-κB) to promote the expression of proinflammatory cytokines, such as tumor necrosis factor (TNF)-α, interleukin-1β, cancer cell growth, cell invasion, and the expression of antiapoptotic protein Bcl-2. PVS might downregulate Bcl-2 by scavenging ROS [47], leading to apoptosis in HPV-infected keratinocytes [16]. In the microarray experiments using samples of EGW patients treated with sinecatechins, the reduced expression of TNF was demonstrated in the responder [45]. EGW tissues that were treated with sinecatechins showed reduced expression of REL, a component of NF-κB [45,48]. These findings suggest the involvement of downregulation of the NF-κB pathway in EGW clearance that is induced by PSV.

In harmony with these actions of EGCG, PSV was demonstrated to downregulate angiogenesis [49], activator protein-1 (AP-1) [50,51,52], ROS [53], NF-κB [54], TNF [55], Akt/PI3K [50], EGFR [50,56], ERK1/2 [50,52,56], cell cycle progression [41,50,57], vascular endothelial growth factor (VEGF) [58], inflammation [52,54], and to upregulate apoptosis [41,50,59] and Bak [16] (Figure 2). Thus, the effects elicited by PSV appear to be mainly attributable to those of its component EGCG.

In addition, EGCG can induce apoptosis through binding to cell surface protein 67kDa laminin receptor [40]. The binding of EGCG to matrix metalloproteinases (MMP) would result in the inhibition of their activities that are involved in invasion and metastasis of cancer cells [60]. The inhibition of MMPs, including MMP-2 and MMP-9, has been suggested in EGW clearance [56]. It remains to be explored to understand how these protein binding properties of EGCG and PSV are linked to their effects on the diseases discussed here.

Several papers have described the involvement of modulation of cellular immunity in the clearance of GW [16,42,43]. Polyphenon^®^E was found to inhibit tumor growth by acting on myeloid-derived suppressor cells and CD8^+^ T cells, indicating immune activation [61], which is consistent with the infiltration of activated T cells and upregulation of pro-inflammatory cytokines in spontaneously regressed EGW [39]. In contrast, Zhang et al. [62] demonstrated that EGCG attenuated imiquimod-induced skin inflammation, as evidenced by, for example, a reduction of infiltration of T cells in the dermal lesions and the decreased plasma levels of IL-17 and IL-23 in accordance with the known immunosuppressive action of EGCG [55,63,64]. In addition, EGCG was found to cause an elevation of CD4^+^ T-cells’ population without affecting the population of CD8^+^ T-cells in the spleen [62]. These inconsistent findings indicate that future studies are required to understand how immunomodulating action of PSV is associated with its effects on diseases that are caused by HPV infection.

## Figures and Tables

**Figure 1 molecules-25-02588-f001:**
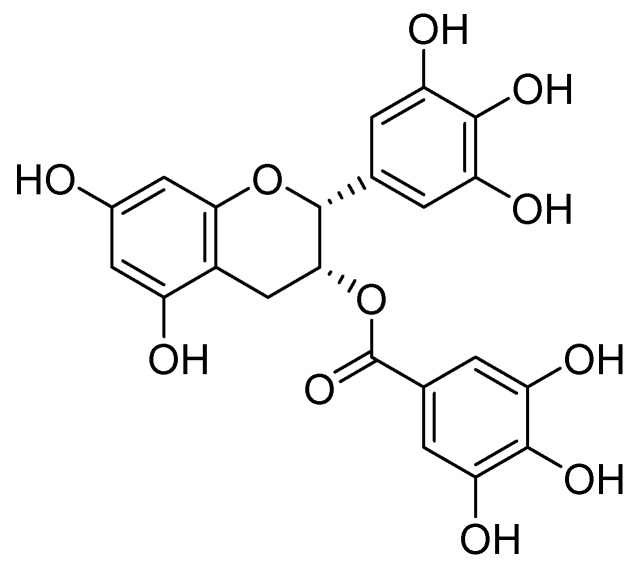
Chemical structure of (−)-epigallocatechin gallate (EGCG).

**Figure 2 molecules-25-02588-f002:**
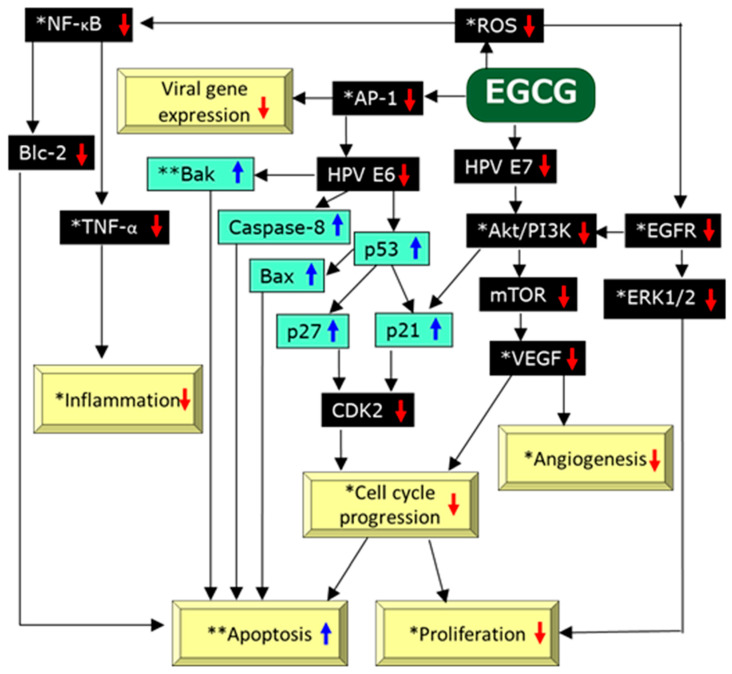
Possible mechanisms of actions of EGCG on HPV-infected cells adopted from literatures [16,40,47]. PSV was shown to * down-regulate angiogenesis [49], AP-1 [50,51,52], ROS [53], NF-κB [54], TNF [55], Akt/PI3K [50], EGFR [50,56], ERK1/2 [50.52,56], cell cycle progression [41,50,57], VEGF [58], inflammation [52,54], and to ** upregulate apoptosis [41,50,59] and Bak [16].

**Table 1 molecules-25-02588-t001:** Case reports for effects of Polyphenon^®^E/sinecatechins/Veregen^®^ (PSV) on viral warts.

Case	PSV Application	Subject	Major Outcome	Reference
1	SC 10%	34-year old male with recalcitrant facial warts and atopic dermatitis	Complete remission within 20 days with light skin irritation	[23]
2	PE 10%3 times/day	An HIV-positive 55-year-old male with 5 GW lesions	Reduction to 3 lesions after one month and complete regression after 8 weeks	[24]
3	SC 10%	HIV-positive female with previous unsuccessful treatments of cryotherapy plus imiquimod	Complete regression of inflammation and reduced number of GW lesion after 3 weeks No recurrence during 8 months	[25]
4	SC 10%	11-year-old child with AGW	Complete regression of the lesions after 10 weeks No notable side effects during a 12 week-follow-up	[26]
5	* PE 10%, twice/day	67-year-old male with plantar warts with unsuccessful prior treatments including cryotherapy and 5-fluorouracil regimen	Complete regression after 3 months	[27]

* PE—Polyphenon^®^E; SC—sinecatechins.

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
