# Peer review of "Applications of a Standardized Green Tea Catechin Preparation for Viral Warts and Human Papilloma Virus-Related and Unrelated Cancers"

_molecules, 2020, doi:10.3390/molecules25112588_

Round 1
Reviewer 1 Report
The Authors describe the biological properties of EGCG from Green Tea, discussing the different applications for human health. In particular they enlighten the clinical efficacy demonstrated in the treatment of genital warts and in other HPV-related diseases.
The manuscript is well developed and described; references are adequate and updated.
I only suggest the addition of these two references in the introduction of the main text
1) Narotzki, B.; Reznick, A.Z.; Aizenbud, D.; Levy, Y. Green tea: a promising natural product in oral health. 430 Archives of oral biology 2012, 57, 429-435, doi:10.1016/j.archoralbio.2011.11.017.
2) Chacko, S.M.; Thambi, P.T.; Kuttan, R.; Nishigaki, I. Beneficial effects of green tea: a literature review. 436 Chinese medicine 2010, 5, 13, doi:10.1186/1749-8546-5-13.
Author Response
The Authors describe the biological properties of EGCG from Green Tea, discussing the different applications for human health. In particular they enlighten the clinical efficacy demonstrated in the treatment of genital warts and in other HPV-related diseases.
The manuscript is well developed and described; references are adequate and updated.
I only suggest the addition of these two references in the introduction of the main text
1) Narotzki, B.; Reznick, A.Z.; Aizenbud, D.; Levy, Y. Green tea: a promising natural product in oral health. 430 Archives of oral biology 2012, 57, 429-435, doi:10.1016/j.archoralbio.2011.11.017.
2) Chacko, S.M.; Thambi, P.T.; Kuttan, R.; Nishigaki, I. Beneficial effects of green tea: a literature review. 436 Chinese medicine 2010, 5, 13, doi:10.1186/1749-8546-5-13.
Thank you very much for your suggestion. We have newly added suggested two references in the revised manuscript as Reference [2] and [3].
Reviewer 2 Report
This manuscript reviews the effect of EGCG (the main component of PSV) on warts. It is therefore limited to one drug (a mixture of EGCG and other catechins) to one problem (or to other HPV related diseases). The title does not correspond to the article as it can be seen when reading the abstract.
Nevertheless, the article is well written and organized but my opinion is that such review cannot find a large public.
Author Response
This manuscript reviews the effect of EGCG (the main component of PSV) on warts. It is therefore limited to one drug (a mixture of EGCG and other catechins) to one problem (or to other HPV related diseases). The title does not correspond to the article as it can be seen when reading the abstract.
Nevertheless, the article is well written and organized but my opinion is that such review cannot find a large public.
Thank you for your constructive suggestion. Other reviewers also indicated the same point. According to your suggestion, the title has been revised: Applications of a Standardized Green Tea Catechin Preparation for Viral Warts and Human Papilloma Virus-Related and Unrelated Cancers
Reviewer 3 Report
The review summarized the effect of Polyphenon®E/sinecatechins/Veregen® 20 (PSV), an FDA approved standardized green tea catechins preparation, on HPV-related warts/cancers and other HPV unrelated cancers such as breast cancer. It also reported how (-)-epigallocatechin gallate (EGCG), the major component of PSV, works to induce apoptosis and exert anti-inflammatory actions in HPV-related diseases.
However, the review is not organized to be easily followed even with the subheadings. The title is too broad for the content. “Applications of Green Tea Catechins for HPV-related and unrelated cancers” would sound better. While the most part of the review is focusing on HPV-associated diseases and cancers, the authors mentioned about the effect of PSV on other types of tumors/cancers (Line 146-193: chronic lymphocytic leukemia, ulcerative colitis, bladder tumor, breast cancer, multiple sclerosis).
The authors could combine the case studies into the section.
Author Response
The review summarized the effect of Polyphenon®E/sinecatechins/Veregen® 20 (PSV), an FDA approved standardized green tea catechins preparation, on HPV-related warts/cancers and other HPV unrelated cancers such as breast cancer. It also reported how (-)-epigallocatechin gallate (EGCG), the major component of PSV, works to induce apoptosis and exert anti-inflammatory actions in HPV-related diseases.
However, the review is not organized to be easily followed even with the subheadings. The title is too broad for the content. “Applications of Green Tea Catechins for HPV-related and unrelated cancers” would sound better. While the most part of the review is focusing on HPV-associated diseases and cancers, the authors mentioned about the effect of PSV on other types of tumors/cancers (Line 146-193: chronic lymphocytic leukemia, ulcerative colitis, bladder tumor, breast cancer, multiple sclerosis).
The authors could combine the case studies into the section.
Thank you for your important suggestions. The title has been revised by incorporating your suggested title. New title is: Applications of a Standardized Green Tea Catechin Preparation for Viral Warts and Human Papilloma Virus-Related and Unrelated Cancers
A section “5. Epidemiological studies of PSV effects on HPV-unrelated cancers has been created. We have also combined Case reports into one section “4. Case reports of PSV effects on viral warts and other HPV-related diseases”
Reviewer 4 Report
The manuscript “Applications of Green Tea Catechins for Human Health” concerns a very actually issue of phyto-therapeutic applications of green tea catechins.
The Authors provide literature data from which they conclude that the green tea catechins may support the treatment of various diseases.
The structure of the manuscript is very good. However, there are a several points that need correction:
- The title of the manuscript may be misleading, because the Authors described mainly the usefulness of green tea catechins in HPV-related entities, especially in a various types of warts. Thus, I suggest to change the title as more applicable to the main text. Alternatively, the Authors may modified the organization of main text by separate some sections regarding different diseases (like genital warts, cancers, neurodegenerative disorders etc.).
- The paragraph about the possible side effects of local as well systemic green tea catechins treatment should be added.
- Figure 1 should be relocated directly in the Introduction section, where it is cited (according to Instructions for Authors).
- Table 1 should be relocated directly in the “Case studies of PSV effects on viral warts” section, where it is cited.
- The number of reference should be introduced in the main text after the name of cited author (e.g. page 3 line 99, page 3 line 110, page 3 line 127, page 4 line 158, page 4 line 180, page 4 line 195).
After providing these remarks, this paper will be suitable for publication in „Molecules”.
Author Response
The manuscript “Applications of Green Tea Catechins for Human Health” concerns a very actually issue of phyto-therapeutic applications of green tea catechins.
The Authors provide literature data from which they conclude that the green tea catechins may support the treatment of various diseases.
The structure of the manuscript is very good. However, there are a several points that need correction:
1. The title of the manuscript may be misleading, because the Authors described mainly the usefulness of green tea catechins in HPV-related entities, especially in a various types of warts. Thus, I suggest to change the title as more applicable to the main text. Alternatively, the Authors may modified the organization of main text by separate some sections regarding different diseases (like genital warts, cancers, neurodegenerative disorders etc.).
Thank you for your precious suggestions.
As for the title, other reviewers also pointed out the same. New title is: Applications of a Standardized Green Tea Catechin Preparation for Viral Warts and Human Papilloma Virus-Related and Unrelated Cancers
We have modified organization. Sections (5. Epidemiological studies of PSV effects on HPV-unrelated cancers, 6. Clinical studies of PSV effects on other diseases and 7. Adverse effects of PSV) have been created and some descriptions have been moved to the corresponding section.
2. The paragraph about the possible side effects of local as well systemic green tea catechins treatment should be added.
As suggested, the separate section of “7. Adverse effects of PSV” has been added.
3. Figure 1 should be relocated directly in the Introduction section, where it is cited (according to Instructions for Authors).
As suggested, Fig. 1 has been relocated according to the Journal’s instruction.
4. Table 1 should be relocated directly in the “Case studies of PSV effects on viral warts” section, where it is cited.
Table 1 has been relocated as suggested,
5. The number of reference should be introduced in the main text after the name of cited author (e.g. page 3 line 99, page 3 line 110, page 3 line 127, page 4 line 158, page 4 line 180, page 4 line 195).
As suggested, numbers of references have been relocated.
Round 2
Reviewer 3 Report
The authors have made progress in organizing the manuscript. However, several additional changes will further increase the readability for the audience.
The author provided studies supporting the efficacy of PSV and also cases that no effect was found in some studies. Many readers would be perplexed about whether PSV really works or not. It would be very helpful if the authors discuss about the conflict results and offer some explanation why this happened. For example, differences in the formulation and treatment regime as well as different population of patients may contribute to the conflict results.
The authors need to better define several terms in a paragraph and clarify their relations with HPV infection, these include EGW, GW, AGW, VW, cervical cancer etc. The authors should be able to locate abundant information on HPV associated warts and cancers.
Line 51-52 the sentence is confusing
Line 70-84 the detail of these three studies would be more appropriate in section 2, the summary of these studies could be included in the introduction.
Line 97 2. Epidemiological studies of PSV effects on GW? The content of this section is about “the effective of sinecatechins on GW and in proactive sequential therapy after ablative strategies”
Line 125 3. Epidemiological studies of PSV effects on other HPV-related diseases? From the content, the title would be “Conflict results of PSV effects on other HPV-related diseases”
Line 165-171 Is that case a GW case by HPV infection? If it is it should be relocated to section 2
Line 182 5. Epidemiological studies of PSV effects on HPV-unrelated cancers “Epidemiological studies of” can be eliminated
Author Response
Thank you for valuable suggestions of reviewer 3. We revised largely according to these comments. Author’s responses (Au) are as follows.
Comments and Suggestions for Authors
The authors have made progress in organizing the manuscript. However, several additional changes will further increase the readability for the audience.
1) The author provided studies supporting the efficacy of PSV and also cases that no effect was found in some studies. Many readers would be perplexed about whether PSV really works or not. It would be very helpful if the authors discuss about the conflict results and offer some explanation why this happened. For example, differences in the formulation and treatment regime as well as different population of patients may contribute to the conflict results.
Au: According to the suggestion, we revised section 7 which has now discussion on inconsistency among the study results as below. New title is: 7. Inconsistency among study results and adverse effects of PSV.
As discussed above, a number of results from clinical studies have provided evidence to show efficacy of PSV on viral warts and some other diseases. However, several studies have failed to show such favorable effects [30,34,37,38]. These inconsistent results may have arisen from potential confounding factors including differences in disease state at drug application, infected virus types, drug formulations, treatment methods, genetic factors, and life styles such as cigarette smoking and alcohol drinking. Thus, future high quality randomized clinical trials would reveal the efficacy of PSV.
2) The authors need to better define several terms in a paragraph and clarify their relations with HPV infection, these include EGW, GW, AGW, VW, cervical cancer etc. The authors should be able to locate abundant information on HPV associated warts and cancers.
Au: Definition of these diseases was made by respective authors based on their diagnosis which we should rely on, so it is difficult for us to define individual disease term and clarify their relations with HPV infection.
3) Line 51-52 the sentence is confusing
Au: We revised this sentence.
4) Line 70-84 the detail of these three studies would be more appropriate in section 2, the summary of these studies could be included in the introduction.
Au: According to the suggestion, these paragraphs were moved to section 2 and these references were cited in Introduction.
5) Line 97 2. Epidemiological studies of PSV effects on GW? The content of this section is about “the effective of sinecatechins on GW and in proactive sequential therapy after ablative strategies”
Au: The section title was changed to “2. The efficacy of PSV on GW and proactive sequential therapy after ablative treatments”.
- Line 125 3. Epidemiological studies of PSV effects on other HPV-related diseases? From the content, the title would be “Conflict results of PSV effects on other HPV-related diseases”
Au: The section title was changed to “3. PSV effects on other HPV-related diseases”. Discussion on conflicting results is now seen in the section: “7. Inconsistency among study results and adverse effects of PSV”.
7) Line 165-171 Is that case a GW case by HPV infection? If it is it should be relocated to section 2.
Au: Vulvar squamous cell carcinoma is a kind of HPV-associated female genital cancer [27329249]. Accordingly, we have left as it is.
- Line 182 5. Epidemiological studies of PSV effects on HPV-unrelated cancers “Epidemiological studies of” can be eliminated.
Au: “Epidemiological studies of” was eliminated. New title is: 5. PSV effects on HPV-unrelated cancers